# Analysis of mobility level of COVID-19 patients undergoing mechanical ventilation support: A single center, retrospective cohort study

**Ricardo Kenji Nawa**[1]⊚*, **Ary Serpa Neto**[1,2,3]⊚, **Ana Carolina Lazarin**[1‡], **Ana Kelen da Silva**[1‡], **Camila Nascimento**[1‡], **Thais Dias Midega**[1‡], **Raquel Afonso Caserta Eid**[1‡], **Thiago Domingos Corrêa**[1]⊚, **Karina Tavares Timenetsky**[1]⊚

**1** Department of Critical Care Medicine, Hospital Israelita Albert Einstein, São Paulo, SP, Brazil, **2** Australian and New Zealand Intensive Care-Research Centre (ANZIC-RC), Monash University, Melbourne, Australia, **3** Data Analytics Research & Evaluation (DARE) Centre, Austin Hospital and University of Melbourne, Melbourne, Victoria, Australia

⊚ These authors contributed equally to this work.
‡ ACL, AKS, CN, TDM and RACE also contributed equally to this work.
* ricardo.nawa@einstein.br

**Data Availability Statement:** All relevant data are within the paper and its Supporting Information file.

## Abstract

### Background

Severe coronavirus disease 2019 (COVID-19) patients frequently require mechanical ventilation (MV) and undergo prolonged periods of bed rest with restriction of activities during the intensive care unit (ICU) stay. Our aim was to address the degree of mobilization in critically ill patients with COVID-19 undergoing to MV support.

### Methods

Retrospective single-center cohort study. We analyzed patients' mobility level, through the Perme ICU Mobility Score (Perme Score) of COVID-19 patients admitted to the ICU. The Perme Mobility Index (PMI) was calculated [PMI = ΔPerme Score (*ICU discharge–ICU admission*)/ICU length of stay], and patients were categorized as "improved" (PMI > 0) or "not improved" (PMI ≤ 0). Comparisons were performed with stratification according to the use of MV support.

### Results

From February 2020, to February 2021, 1,297 patients with COVID-19 were admitted to the ICU and assessed for eligibility. Out of those, 949 patients were included in the study [524 (55.2%) were classified as "improved" and 425 (44.8%) as "not improved"], and 396 (41.7%) received MV during ICU stay. The overall rate of patients out of bed and able to walk ≥ 30 meters at ICU discharge were, respectively, 526 (63.3%) and 170 (20.5%). After adjusting for confounders, independent predictors of improvement of mobility level were frailty (OR: 0.52; 95% CI: 0.29–0.94; *p* = 0.03); SAPS III Score (OR: 0.75; 95% CI: 0.57–0.99; *p* = 0.04); SOFA Score (OR: 0.58; 95% CI: 0.43–0.78; *p* < 0.001); use of MV after the first hour of ICU admission (OR: 0.41; 95% CI: 0.17–0.99; *p* = 0.04); tracheostomy (OR: 0.54; 95% CI: 0.30–

**Funding:** The author(s) received no specific funding for this work.

**Competing interests:** The authors have declared that no competing interests exist.

0.95; $p = 0.03$); use of extracorporeal membrane oxygenation (OR: 0.21; 95% CI: 0.05–0.8; $p = 0.03$); neuromuscular blockade (OR: 0.53; 95% CI: 0.3–0.95; $p = 0.03$); a higher Perme Score at admission (OR: 0.35; 95% CI: 0.28–0.43; $p < 0.001$); palliative care (OR: 0.05; 95% CI: 0.01–0.16; $p < 0.001$); and a longer ICU stay (OR: 0.79; 95% CI: 0.61–0.97; $p = 0.04$) were associated with a lower chance of mobility improvement, while non-invasive ventilation within the first hour of ICU admission and after the first hour of ICU admission (OR: 2.45; 95% CI: 1.59–3.81; $p < 0.001$) and (OR: 2.25; 95% CI: 1.56–3.26; $p < 0.001$), respectively; and vasopressor use (OR: 2.39; 95% CI: 1.07–5.5; $p = 0.03$) were associated with a higher chance of mobility improvement.

## Conclusion

The use of MV reduced mobility status in less than half of critically ill COVID-19 patients.

## Introduction

The 2019 novel coronavirus severe acute respiratory syndrome coronavirus 2 (SARS-CoV-2) was first identified and reported in Wuhan, Hubei province, China, in December 2019, as the cause of a respiratory illness designated as coronavirus disease 2019 (COVID-19) [1]. Since then, the COVID-19 has infected over 551 million individuals globally and decimated over 6.35 million lives worldwide [2].

Severe cases of COVID-19 patients frequently develop respiratory failure along with extra-pulmonary organ dysfunctions, requiring prolonged periods of life support and hospitalization in the intensive care unit (ICU) [3, 4]. Up to one-third of hospitalized patients with COVID-19 received invasive mechanical ventilation (MV) due to severe pneumonia [5–9]. Patients undergoing MV frequently require deep sedation, infusion of neuromuscular blocking agents (NMBAs), and have poor prognosis, with a high risk of death [10–12].

Despite the use of MV support, another important risk factor associated with worse clinical outcomes in ICU patients with COVID-19 is prolonged bed rest. One of the consequences of prolonged ICU stay and/or immobility is the high risk of development of ICU-acquired weakness (ICU-AW) due to muscle loss [13, 14]. Indeed, up to 66% of hospitalized patients will be diagnosed with ICU-AW, leading to deficits and/or impairments in physical function [15, 16]. The rate of skeletal muscle wasting occurs early in patients with acute respiratory distress syndrome (ARDS) and multiple organ failure diagnosis [17]. For instance, a decrease of up to 10–20% of the muscles of quadricep complex cross-sectional area within 7 to 10 days in patients diagnosed with ARDS and multiple organ failure has been demonstrated [15]. The consequences are frequently observed among ICU survivors, with impairment in physical and psychological recovery [18]. Early mobilization and rehabilitation have been shown to be effective in enhancing the recovery of critically ill patients, but more large-scale, multicenter randomized controlled trials are required to further confirm these findings [19]. Although the present available evidence of early rehabilitation still remains inconsistent; on the overall evidence rehabilitation interventions should not be delayed [20].

The present study analyzes the degree of mobilization in critically ill patients with COVID-19. Our group has completed a previous study analyzing the assessment of mobility level in patients with COVID-19 [21], but at present, no data are available describing a similar degree of mobilization and outcomes analysis in critically ill COVID-19 population admitted to the

ICU. We hypothesized that variations in clinical characteristics, use of MV, risk factors associated with mobility level, and resource use were related to evolving changes in mobility during ICU stay.

## Methods

### Study design

This single-center retrospective cohort study was conducted in a private quaternary hospital located in the city of São Paulo, Brazil. The Hospital Israelita Albert Einstein comprises a total of 634 beds. The study was approved by the Institutional Review Board (IRB) of Hospital Israelita Albert Einstein's ethics committee under number CAAE: 30797520.6.0000.0071 and informed consent was waived. This study is reported in accordance with the Strengthening the Reporting of Observational studies in Epidemiology (STROBE) statement [22].

### Patients

We considered all ICU admissions of COVID-19 diagnosis (e.g., mild, moderate, and severe) during the study period eligible for inclusion. The following inclusion criteria were used: 1) age $\geq$ 18 years; and 2) confirmed diagnosis of COVID-19 by reverse transcription–polymerase chain reaction (RT-PCR) for SARS-CoV-2 [23]. We excluded patients with missing core data, defined as the use of MV during ICU stay and report of mobility status at ICU admission and/or discharge.

### Data collection and study variables

All study data were retrieved from the electronic medical record (EMR) and the Epimed Monitor System® (Epimed Solutions, Rio de Janeiro, Brazil), which is an electronic structured case report form where patients' data are prospectively entered by trained ICU case managers [24]. The EMR was accessed between February 1, 2020 and February 28, 2021. All data were extracted by an independent research assistant that did not participate in this study. Data were fully anonymized prior to being made available to researchers.

Collected variables included demographics, comorbidities, Simplified Acute Physiology Score (SAPS III score) at ICU admission–scores ranging from 0 to 217, with higher scores indicating more severe illness and higher risk of death [25], Sequential Organ Failure Assessment score (SOFA score) at ICU admission–scores ranging from 0 to 4 for each organ system, with higher aggregate scores indicating more severe organ dysfunction [26], Charlson Comorbidity Index–range from 0 to 5 for each comorbidity, with score of zero indicating that no comorbidities were found. The higher the score, the more likely the predicted outcome will result in mortality or higher resource use [27], Modified Frailty Index–categorized frailty using MFI values into non-frail (MFI = 0), pre-frail (MFI = 1–2) or frail (MFI $\geq$ 3) [28], resource use and organ support [vasopressors, neuromuscular blocking agents (NMBAs), MV, noninvasive ventilation (NIV), renal replacement therapy (RRT), and extracorporeal membrane oxygenation (ECMO)] at ICU admission and during ICU stay, need for tracheostomy, duration of MV, ICU and hospital length of stay (LOS), and ICU and in-hospital mortality.

### Mobility status assessment

All consecutive patients admitted to the ICU with a confirmed diagnosis of COVID-19 who were assessed by a physical therapist had their mobility status evaluated daily, from ICU admission to discharge, with the Perme Intensive Care Unit Mobility Score (Perme Score) [29]. The Perme Score was specifically developed to assess the mobility status of patients

admitted to the ICU [29]. The total score ranges from 0 to 32 points, with higher scores indicating higher mobility status [29]. The Perme Mobility Index (PMI) was also calculated by the difference between the total Perme Score at ICU discharge and the total Perme Score at ICU admission, divided by the ICU length of stay (ICU LOS), as follows: [PMI = ΔPerme Score (*ICU discharge–ICU admission*) / ICU LOS] [21].

## Standard physiotherapy care

According to our institution's early mobilization protocol, all COVID-19 patients admitted to the ICU are assessed by the physiotherapy team for an initial evaluation. Afterwards, all patients are daily evaluated by a physical therapist. The Perme Score is part of the daily mobility status evaluation in the early mobility protocol. Due to the need for isolation in COVID-19 patients, therapies were performed only around the ICU beds. Therefore, all ICU beds are individually isolated, with enough space to perform out of bed exercises (about 82 square feet) while maintaining isolation during therapy.

## Outcomes

The primary outcome was the improvement in mobility, defined as a PMI > 0. Secondary outcomes included key elements of mobilization, defined as being out of bed (and the time until the event) and able to walk at least one meter (and the time until the event). Additional secondary clinical outcomes included duration of MV, ICU and hospital length of stay, ICU and hospital mortality, and 28-day in-hospital mortality.

## Statistical analysis

All patients included in the period who fulfilled inclusion criteria and did not meet any exclusion criterion were included. Continuous variables are presented as median and interquartile range (IQR), and categorical variables as absolute and relative frequencies. Normality was assessed by the Kolmogorov-Smirnov test. Patients were classified as "improved" (PMI > 0) or "not improved" (PMI ≤ 0) and all analyses reported are stratified according to the use of MV during ICU stay. Categorical variables were compared using Fisher exact test, and continuous variables were compared using Wilcoxon rank-sum test. Since the exposure variable is highly correlated with outcomes, no direct assessment of the exposure with the outcome was done (instead of) besides simple comparisons made between the groups.

A multivariable logistic regression model was used to identify factors independently associated with improvement in mobility. A list of candidate baseline predictors was determined a priori and it included only variables with known or suspected relationship with outcome. The multivariable model was constructed considering variables with a *p* < 0.05 in the univariable analysis. Multicollinearity in the final model was assessed using variance-inflation factors, and linearity assumption of continuous variables was assessed using Box-Tidwell transformation considering the full model, testing the log-odds and the predictor variable. Odds ratios and their respective 95% confidence intervals (OR, 95% CI) are reported. All continuous variables were entered after standardization and the OR represents the increase in one standard deviation of the variables.

Key elements of mobilization were reported according to predefined clinical characteristics, defined as: 1) use of MV (*yes* or *no*); 2) age (< 65 vs ≥ 65 years); 3) median SAPS III (< 50 vs. ≥ 50); 4) median Charlson comorbidity score (< 1 vs. ≥ 1); 5) body mass index (< 25 vs. 25–30 vs. > 30 kg/m$^2$); and 6) MFI (non-frail vs. pre-frail vs. frail). The time until the event is presented in Kaplan-Meier curves and compared using unadjusted Cox proportional hazard models. To account for the competing risk of death, patients who died without achieving the

event of interest were assigned the worst time possible. The rate of missing data was low (**S1 Table**) and missing data in predictors were imputed by median. All analyses were conducted in R Version 4.0.3 (R Foundation) [30] and significance level was set at 0.05.

## Results

### Patients

From February 2020 to February 2021, 1,297 patients with confirmed COVID-19 were admitted to the ICU, of which a total of 949 (73.1%) met the inclusion criteria and were included in subsequent analysis. All 348 patients excluded were excluded due to missing data in PMI. From 949 patients studied, 524 (55.2%) were classified as "Improved PMI" and 425 (44.8%) as "Not improved PMI". In addition, 396 (41.7%) received MV during ICU stay, while 553 (58.3%) did not receive it. Baseline characteristics of pooled patients according to the pre-specified groups and stratified by the use of MV are shown in Table 1.

The median (IQR) age of patients was 67 (55–77) years, 68.1% were male, the median SAPS III and SOFA were, respectively, 50 (43–57) and 2 (0–6), and 48.7% were in the pre-frail state. The most prevalent comorbidity was diabetes (36.9%), an overall sample median (IQR) body mass index (BMI: 25+ Kg/m$^2$, classified as overweight) was 27.9 (25.0–31.0), and 10.4% received MV and 7.5% vasopressors within 1 hour of ICU admission. The overall ICU and hospital mortality were 14.9% and 16.2%, respectively (Table 1).

Among patients receiving MV during ICU stay, patients who improved mobility were younger, had lower SAPS III, SOFA, and Charlson comorbidity score, and were less often frail (Table 1). Importantly, patients who improved more often received MV and vasopressor within 1 hour of ICU admission. A similar pattern was found in patients not receiving ventilation during ICU stay; however, patients who improved more often received NIV within 1 hour of ICU admission. Additional organ support during ICU stay is reported in **S2 Table**.

### Mobility levels during ICU stay

The median PMI in the overall study cohort was 0.2 (-0.2–1.1). PMI was lower in patients under MV compared to patients who did not receive MV (0.0 [-0.1–0.7] vs. 0.3 [-0.4–1.8]; *p* = 0.017) presented in **S3 Table**. Among patients under MV, patients who improved mobility during ICU stay had lower Perme Score at admission compared with patients who did not improve (Fig 1 and **S3 Table**). Perme Score became higher in patients who improved after day 5 of follow-up and stayed higher until day 28. In patients not receiving MV, Perme Score became higher in the improved group after day 3 of follow-up, but the difference became small after day 9 (Fig 1 and **S3 Table**). Perme Score at discharge is shown in **S1 Fig**. The improvement in mobility at discharge according to ICU LOS and in the specific subgroups is shown in **S2** and **S3** **Figs**.

### Factors associated with improved mobility

The univariable assessments of factors associated with improvement in mobility are shown in Table 2. After adjustment for confounders, a higher severity of the disease and organ dysfunction (as measured by SAPS III and SOFA), presence of frailty, limitation of therapy orders, use of MV after the first hour of ICU admission, presence of tracheostomy, use of ECMO and NMBAs, a higher Perme Score at admission, and a longer ICU LOS were all associated with a lower chance of improvement in mobility. The use of NIV and the use of vasopressor were associated with a higher chance of improvement in mobility. Additional diagnostics tests of the model are shown in **S4 Table**.

**Table 1. Baseline characteristics and clinical outcomes of the included patients.**

| | Overall (n = 949) | Mechanical Ventilation (n = 396) | | | No Mechanical Ventilation (n = 553) | | |
|---|---|---|---|---|---|---|---|
| | | Improved PMI (n = 202) | Not Improved PMI (n = 194) | p value | Improved PMI (n = 322) | Not Improved PMI (n = 231) | p value |
| Age, years | 67 [55–77] | 60 [50–72] | 74 (63–84) | <0.001* | 64 [52–75] | 68 [55–82] | 0.005* |
| Male gender–no., % | 646 (68.1) | 143 (70.8) | 127 (65.5) | 0.28 | 222 (68.9) | 154 (66.7) | 0.58 |
| Body mass index[a], kg/m$^2$ | 27.9 [25–31] | 28.4 [25.9–32.8] | 27.8 [25.1–30.8] | 0.07 | 28 [25.4–30.7] | 27 [24.4–30.5] | 0.04* |
| Severity of illness | | | | | | | |
| SAPS III score[b] | 50 [43–57] | 51 [46–58] | 58 [52–65] | <0.001* | 46 [42–52] | 47 [42–55] | 0.003* |
| SOFA score[c] | 2 [0–6] | 6 [3–7] | 7 [5–9] | <0.001* | 1 [0–2] | 1 [0–2] | 0.38 |
| Hours between hospital and ICU admission | 1 [0–2] | 0 [0–2] | 0 [0–2] | 0.36 | 1 [0–3] | 1 [0–2] | 0.21 |
| Charlson comorbidity score[d] | 1 [0–2] | 0 [0–1] | 1 [0–3] | <0.001* | 0 [0–1] | 1 [0–2] | 0.02* |
| Modified frailty score | 1 [0–2] | 1 [0–2] | 2 [1–3] | <0.001* | 1 [0–2] | 1 [0–2] | 0.04* |
| Non-frail–no., % | 327 (34.5) | 89 (44.1) | 22 (11.3) | | 131 (40.7) | 85 (36.8) | |
| Pre-frail–no., % | 462 (48.7) | 94 (46.5) | 117 (60.3) | <0.001* | 152 (47.2) | 99 (42.9) | 0.03* |
| Frail–no. (%) | 160 (16.9) | 19 (9.4) | 55 (28.4) | | 39 (12.1) | 47 (20.3) | |
| Readmission–no., % | 6 (0.6) | 2 (1) | 0 (0) | 0.49 | 2 (0.6) | 2 (0.9) | 0.99 |
| Co-morbidities–no., % | | | | | | | |
| Diabetes | 280 (36.9) | 53 (36.1) | 83 (45.1) | 0.11 | 90 (36.3) | 54 (30) | 0.18 |
| Solid neoplasia | 77 (10.1) | 14 (9.5) | 22 (12) | 0.59 | 23 (9.3) | 18 (10) | 0.86 |
| COPD | 64 (8.4) | 9 (6.1) | 23 (12.5) | 0.06 | 16 (6.5) | 16 (8.9) | 0.35 |
| Heart failure | 59 (7.8) | 6 (4.1) | 21 (11.4) | 0.01* | 12 (4.8) | 20 (11.1) | 0.02* |
| Chronic kidney disease | 55 (7.2) | 10 (6.8) | 23 (12.5) | 0.09 | 9 (3.6) | 13 (7.2) | 0.12 |
| Previous myocardial infarction | 35 (4.6) | 5 (3.4) | 12 (6.5) | 0.22 | 12 (4.8) | 6 (3.3) | 0.47 |
| Hematological cancer | 31 (4.1) | 5 (3.4) | 9 (4.9) | 0.59 | 6 (2.4) | 11 (6.1) | 0.07* |
| Metastatic | 17 (2.2) | 3 (2) | 5 (2.7) | 1 | 5 (2) | 4 (2.2) | 0.99 |
| Hemodialysis | 11 (1.4) | 1 (0.7) | 6 (3.3) | 0.13 | 1 (0.4) | 3 (1.7) | 0.31 |
| Cirrhosis | 2 (0.3) | 1 (0.7) | 0 (0) | 0.44 | 1 (0.4) | 0 (0) | 0.99 |
| Immunosuppression | 1 (0.1) | | | | 1 (0.4) | 0 (0) | 0.99 |
| Within the first hour of ICU admission | | | | | | | |
| Non-invasive ventilation | 227 (23.9) | 59 (29.2) | 53 (27.3) | 0.73 | 81 (25.2) | 34 (14.7) | 0.003* |
| Invasive mechanical ventilation | 99 (10.4) | 75 (37.1) | 24 (12.4) | <0.001* | 0 (0) | 0 (0) | – |
| Renal replacement therapy | 2 (0.2) | 0 (0) | 2 (1) | 0.23 | 0 (0) | 0 (0) | – |
| Vasopressor | 71 (7.5) | 44 (21.8) | 23 (11.9) | 0.01 | 3 (0.9) | 1 (0.4) | 0.64 |
| Acute kidney injury | 48 (5.1) | 16 (7.9) | 21 (10.8) | 0.38 | 4 (1.2) | 7 (3) | 0.21 |
| Limitation of treatment–no., % | 32 (3.4) | 0 (0) | 7 (3.6) | 0.006* | 3 (0.9) | 22 (9.5) | <0.001* |
| Vital signs within 24 hours of admission | | | | | | | |
| Highest temperature, ˚C | 36.4 [36–37] | 36.6 [36–37.2] | 36.4 [35.9–37] | 0.01* | 36.4 [36–37] | 36.4 [35.9–37] | 0.6 |
| Lowest mean arterial pressure, mmHg | 89 [80–99] | 88 [78–98] | 88 [78–98] | 0.88 | 91 [82–99] | 91 [81–99] | 0.5 |
| Highest heart rate, bpm | 80 [70–90] | 82 [71–92] | 80 [70–92] | 0.4 | 78 [71–89] | 79 [70–88] | 0.5 |
| Pathology within 24 hours of admission | | | | | | | |
| Highest white blood cell count, 10$^9$/L | 8.1 [5.7–11.1] | 8.7 [6.7–11.4] | 7.7 [5.4–11.4] | 0.12 | 7.7 [5.3–11] | 7.9 [5–10.1] | 0.65 |
| Lowest platelet, 10$^9$/L | 195 [154–241] | 220 [164–275] | 171 [130–224] | <0.001* | 194 [158–228] | 195 [154–234] | 0.9 |
| Highest creatinine, mg/dL | 0.98 [0.80–1.27] | 0.98 [0.78–1.19] | 1.19 [0.94–2.12] | 0.002* | 0.9 [0.74–1.04] | 0.96 [0.8–1.17] | 0.09 |

(*Continued*)

**Table 1.** (Continued)

| | Overall (n = 949) | Mechanical Ventilation (n = 396) | | | No Mechanical Ventilation (n = 553) | | |
|---|---|---|---|---|---|---|---|
| | | Improved PMI (n = 202) | Not Improved PMI (n = 194) | p value | Improved PMI (n = 322) | Not Improved PMI (n = 231) | p value |
| pH | 7.42 [7.38–7.46] | 7.41 [7.36–7.44] | 7.42 [7.36–7.46] | 0.38 | 7.46 [7.43–7.47] | 7.42 [7.42–7.45] | 0.02* |
| $PaO_2$ / $FiO_2$, mmHg | 256 [172–338] | 222 [172–300] | 240 [101–325] | 0.65 | 335 [255–418] | 343 [267–424] | 0.93 |
| $PaCO_2$, mmHg | 36 [32–43] | 40 [35–45] | 35 [31–48] | 0.12 | 32 [29–35] | 35 [32–38] | 0.01* |
| Lactate, mmol/L | 1.6 [1.1–2] | 1.6 [1.2–2] | 1.6 [1.1–2] | 0.77 | 1.4 [1.1–2] | 1.5 [1–3.4] | 0.72 |
| Clinical outcomes | | | | | | | |
| Duration of ventilation, days | 12 [6–24] | 9 [5–15] | 15 [8–33] | <0.001* | – | – | – |
| In survivors | 10 [5–18] | 9 [5–15] | 12 [6–27] | 0.05 | – | – | – |
| ICU length of stay, days | 6 [3–12] | 10 [4–16] | 16 [8–29] | <0.001* | 5 [3–8] | 4 [2–7] | 0.002* |
| In survivors | 5 [3–10] | 10 [4–16] | 8.5 [3–18] | 0.68 | 5 [3–8] | 4 [2–6] | <0.001* |
| Hospital length of stay, days | 16 [10–28] | 26 [17–37] | 30 [17–52] | 0.09 | 11 [9–17] | 11 [7–16] | 0.05 |
| In survivors | 15 [10–26] | 27 [17–38] | 37 [27–63] | <0.001* | 11 [9–17] | 11 [7–17] | 0.14 |
| ICU mortality–no., % | 141 (14.9) | 5 (2.5) | 116 (59.8) | <0.001* | 3 (0.9) | 17 (7.4) | <0.001* |
| Hospital mortality–no., % | 152 (16.2) | 5 (2.5) | 123 (65.8) | <0.001* | 3 (0.9) | 21 (9.1) | <0.001* |
| 28-day mortality–no., % | 127 (13.4) | 4 (2) | 100 (51.5) | <0.001* | 3 (0.9) | 20 (8.7) | <0.001* |

Data are median and interquartile range [quartile 25%–quartile 75%] or n (%). Percentages may not total 100 because of rounding.

*Definition of abbreviations*: PMI = perme mobility index; SAPS = simplified acute physiology score; SOFA = sequential organ failure assessment; COPD = chronic obstructive pulmonary disease; ICU = intensive care unit; ECMO = extracorporeal membrane oxygenation; $PaO_2$ = partial pressure of oxygen; $PaCO_2$ = partial pressure of carbon dioxide; $FiO_2$ = fraction of inspired oxygen.

*Statistically significant difference between groups.

[a]The body-mass index (BMI) is calculated by weight in kilograms divided by the square of the height in meters ($Kg/m^2$). The categories are the same for men and women of all body types and ages, as follows: below 18·5 –underweight, 18·5–24·9 –normal or healthy weight, 25·0–29·9 –overweight, and 30·0 and above–obese.

[b]The SAPS III score range from 0 to 217, with higher scores indicating more severe illness and higher risk of death.

[c]SOFA scores range from 0 to 4 for each organ system, with higher aggregate scores indicating more severe organ dysfunction.

[d]Charlson comorbidity index range from 0 to 5 for each comorbidity, with score of zero indicating that no comorbidities were found. The higher the score, the more likely the predicted outcome will result in mortality or higher resource use.

## Key elements of mobilization

The percentage of patients able to get out of bed during ICU stay was lower in patients receiving MV (36.4% vs. 72.0% in patients not receiving ventilation; $p < 0.001$), in older patients (51.2% vs. 64.1% in young patients; $p < 0.001$), in more severe patients (45.2% vs. 69.4% in less severe patients; $p < 0.001$), in more comorbid patients (50.1% vs. 64.1% in less comorbidity; $p < 0.001$), and in frail patients (41.2% vs. 56.5% in pre-frail vs. 65.7% in non-frail; $p < 0.001$), data are presented in Table 3 and **S5 Table**. In addition, the percentage of patients walking > 30 meters during ICU stay followed the same pattern.

Patients not receiving MV, young patients, less severe and with less co-morbidities, and non-frail patients got out of bed sooner (Fig 2). A similar pattern was found for the time until first walking (**S4 Fig**) and the time until first walking of more than 30 meters (**S5 Fig**).

## Discussion

The main finding of this single-center cohort of critically ill COVID-19 patients was that approximately fifty one percent of patients submitted to invasive MV improved their mobility status during the ICU stay. We also observed the main factors associated with a lower chance of improvement in mobility were a higher severity of the disease and organ dysfunction,

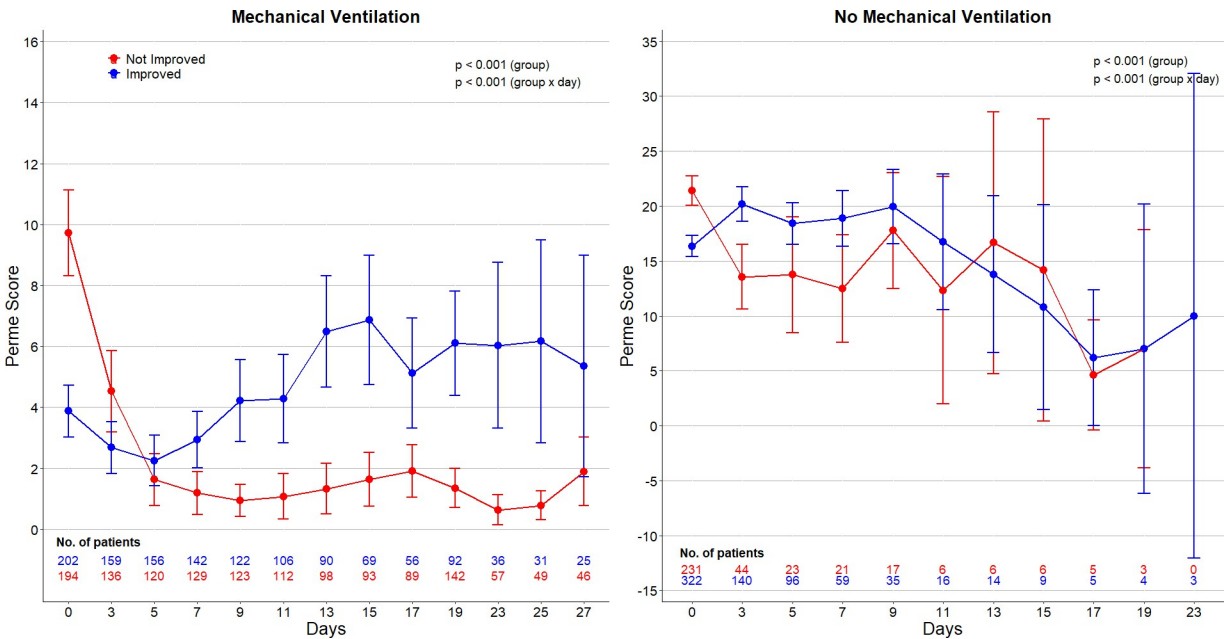

**Fig 1. Perme score over the first 27 days of ICU admission.** Circles are mean and error bars are 95% confidence interval. P value from a mixed-effect generalized linear model with Gaussian distribution, with group and time, and a group x time interaction as fixed effect, and the patients as random effect to account for repeated measurements. The number of patients with available data decreases over successive study days due to deaths and discharges. *Perme ICU mobility score ranges from 0 to 32, with higher scores indicating better mobility level.

presence of frailty, limitation of therapy orders, use of MV after the first hour of ICU admission, presence of tracheostomy, use of ECMO and NMBAs, a higher Perme Score at admission, and a longer ICU stay. The time for the first day out of bed was shorter in patients that did not require MV, who were less severe, younger, non-frail, and with less comorbidity. The use of NIV and the use of vasopressor were associated with a higher chance of improvement in mobility.

COVID-19 patients frequently exhibit long periods of ICU and hospital stay, prolonged periods of MV and higher in-hospital mortality [31, 32]. Moreover, functional outcomes in survivor patients requiring MV are often poor [33]. A study conducted in COVID-19 mechanically ventilated patients reported an incidence of 100% of ICU-AW at the moment of awakening [32]. The mobilization activities started around day 14 after ICU admission, and patients with higher body mass index took longer to be first mobilized [32]. The authors reported a low mobility level at ICU discharge, but patients showed improvement during hospital stay. The mobility level was associated with the clinical frailty score and with previously cardiovascular disease. Less than 10% of patients were able to walk 30 meters or more at ICU discharge [32].

A retrospective single-center cohort study, the mobility level of critically ill COVID-19 patients measured by the Perme Score, was low at ICU admission; however, most patients improved their mobility level during ICU stay [21]. This study also reported the risk factors associated with mobility level were age, comorbidities, and use of renal replacement therapy [21]. In our study, the incidence of mechanically ventilated patients able to perform out-of-bed activities during ICU stay was approximately one third when compared with patients that did not require MV support. While most of the non-ventilated patients were moved out of bed on the first day of ICU admission, in ventilated patients this occurred on the third day only. The impact of MV in mobility level was also observed in previous studies, with patients' mobility being restricted to in-bed exercises [34–37]. Also, MV was reported as a common barrier to

**Table 2. Univariable and multivariate logistic regression analysis addressing risk factors associated with patients' that improved mobility level (n = 949 patients).**

| | Univariable Model | | Multivariable Model* | |
|---|---|---|---|---|
| | OR (95% CI) | p value | OR (95% CI) | p value |
| Age | 0.97 (0.96 to 0.98) | <0.001* | 0.85 (0.66 to 1.08) | 0.18 |
| Male gender | 1.18 (0.89 to 1.55) | 0.24 | – | – |
| SAPS III score | 0.95 (0.94 to 0.97) | <0.001* | 0.75 (0.57 to 0.99) | 0.04* |
| SOFA score | 0.91 (0.87 to 0.94) | <0.001* | 0.58 (0.43 to 0.78) | <0.001* |
| Charlson comorbidity index | 0.82 (0.75 to 0.89) | <0.001* | 1.01 (0.84 to 1.22) | 0.90 |
| Modified frailty score | | | | |
| Non-frail | 1 (Reference) | | 1 (Reference) | |
| Pre-frail | 0.55 (0.41 to 0.74) | <0.001* | 0.79 (0.54 to 1.17) | 0.23 |
| Frail | 0.28 (0.19 to 0.41) | <0.001** | 0.52 (0.29 to 0.94) | 0.03* |
| Palliative care | 0.08 (0.02 to 0.22) | <0.001* | 0.05 (0.01 to 0.16) | <0.001* |
| Organ support | | | | |
| Non-invasive ventilation | | | | |
| No use | 1 (Reference) | | 1 (Reference) | |
| Within 1 hr of admission | 1.86 (1.32 to 2.63) | <0.001* | 2.45 (1.59 to 3.81) | <0.001* |
| After 1 hr of admission | 1.66 (1.24 to 2.23) | 0.001* | 2.25 (1.56 to 3.26) | <0.001* |
| Invasive mechanical ventilation | | | | |
| No use | 1 (Reference) | | 1 (Reference) | |
| Within 1 hr of admission | 2.24 (1.39 to 3.72) | 0.001* | 2.11 (0.65 to 7.09) | 0.21 |
| After 1 hr of admission | 0.54 (0.4 to 0.71) | <0.001 | 0.41 (0.17 to 0.99) | 0.04* |
| Vasopressor | | | | |
| No use | 1 (Reference) | | 1 (Reference) | |
| Within 1 hr of admission | 1.46 (0.88 to 2.49) | 0.15 | 2.22 (0.72 to 6.93) | 0.16 |
| After 1 hr of admission | 0.71 (0.54 to 0.94) | 0.01* | 2.39 (1.07 to 5.5) | 0.03* |
| Renal replacement therapy | 0.31 (0.21 to 0.46) | <0.001* | 0.99 (0.51 to 1.93) | 0.97 |
| Tracheostomy | 0.53 (0.33 to 0.85) | 0.009* | 0.54 (0.30 to 0.95) | 0.03* |
| High-flow nasal cannula | 1.16 (0.89 to 1.51) | 0.28 | – | – |
| ECMO | 0.24 (0.07 to 0.69) | 0.01* | 0.21 (0.05 to 0.8) | 0.03* |
| Use of NMBA | 0.6 (0.46 to 0.79) | <0.001* | 0.53 (0.3 to 0.95) | 0.03* |
| Perme score at ICU admission | 0.96 (0.95 to 0.97) | <0.001* | 0.35 (0.28 to 0.43) | <0.001* |
| ICU length of stay | 0.98 (0.97 to 0.99) | <0.001* | 0.79 (0.61 to 0.97) | 0.04* |

Continuous variables were included after standardization and the odds ratio represents the increase in one standard deviation.

*Definition of abbreviations*: OR = odds ratio; CI = confidence interval; SAPS = simplified acute physiology score; SOFA = sequential organ failure assessment; hr = hour; NMBA = neuromuscular blockade, ECMO = extracorporeal membrane oxygenation; ICU = intensive care unit.

Variables with $p < 0.05$ were selected for the multivariable model.

*Statistically significant.

out-of-bed exercises [36, 37] in accordance with our study as well as a predictor to development of ICU-AW [38, 39].

Sedatives and NMBAs are frequently used in mechanically ventilated patients to promote adequate ventilation [40]. The use of NMBAs was found to be significantly associated with ICU-AW development when administered for 48 hours or more [39, 40]. In our study, approximately 40% of patients required MV support, while three quarters used NMBAs. The need for sedatives and NMBAs induces patients' immobilization, reducing their mobility level during ICU stay and contributing with muscle disuse and, consequently, with ICU-AW [38, 39]. The weakness experienced by survivors of critical illness is thought to be multifactorial,

**Table 3. Degree of mobilization according to different clinical characteristics at baseline.**

| | Overall (n = 949) | Use of Mechanical Ventilation | | | Age (years) | | | SAPS III | | |
|---|---|---|---|---|---|---|---|---|---|---|
| | | Yes (n = 396) | No (n = 553) | p value | ≥65 (n = 514) | <65 (n = 435) | p value | ≥50 (n = 482) | <50 (n = 467) | p value |
| At ICU admission | | | | | | | | | | |
| Out of bed–no., % | 383 (40.4) | 68 (17.2) | 315 (57) | <0.001* | 186 (36.2) | 197 (45.3) | 0.005* | 151 (31.3) | 232 (49.7) | <0.001* |
| Walked any distance–no., % | 180 (19) | 22 (5.6) | 158 (28.6) | <0.001* | 80 (15.6) | 100 (23) | 0.005* | 54 (11.2) | 126 (27) | <0.001* |
| 1–15 meters | 69 (7.3) | 10 (2.5) | 59 (10.7) | | 35 (6.8) | 34 (7.8) | | 27 (5.6) | 42 (9) | |
| 15–30 meters | 32 (3.4) | 7 (1.8) | 25 (4.5) | <0.001* | 16 (3.1) | 16 (3.7) | 0.008* | 12 (2.5) | 20 (4.3) | <0.001* |
| > 30 meters | 79 (8.3) | 5 (1.3) | 74 (13.4) | | 29 (5.6) | 50 (11.5) | | 15 (3.1) | 64 (13.7) | |
| During ICU stay | | | | | | | | | | |
| Out of bed–no., % | 542 (57.1) | 144 (36.4) | 398 (72) | <0.001* | 263 (51.2) | 279 (64.1) | <0.001* | 218 (45.2) | 324 (69.4) | <0.001* |
| Days until first occurrence | 0 [0–3] | 3 [0–11] | 0 [0–0] | <0.001** | 0 [0–3] | 0 [0–3] | 0.95 | 0 [0–3] | 0 [0–3] | 0.23 |
| Walked any distance–no., % | 281 (29.6) | 50 (12.6) | 231 (41.8) | <0.001* | 131 (25.5) | 150 (34.5) | 0.003* | 95 (19.7) | 186 (39.8) | <0.001* |
| Days until first occurrence | 0 [0–3] | 5 [0–13] | 0 [0–3] | <0.001** | 0 [0–4] | 0 [0–3] | 0.36 | 0 [0–5] | 0 [0–3] | 0.04* |
| Walked 1–15 meters–no., % | 124 (13.1) | 27 (6.8) | 97 (17.5) | <0.001* | 70 (13.6) | 54 (12.4) | 0.62 | 52 (10.8) | 72 (15.4) | 0.04* |
| Days until first occurrence | 0 [0–5] | 5 [0–16] | 0 0–3] | <0.001* | 1 [0–5] | 0 [0–4] | 0.35 | 0 [0–5] | 0 [0–5] | 0.50 |
| Walked 15–30 meters–no., % | 72 (7.6) | 21 (5.3) | 51 (9.2) | 0.02* | 34 (6.6) | 38 (8.7) | 0.22 | 29 (6) | 43 (9.2) | 0.06 |
| Days until first occurrence | 3 [0–7] | 11 [0–15] | 3 [0–3] | 0.004* | 3 [0–8] | 3 [0–6] | 0.92 | 3 [0–13] | 3 [0–3] | 0.09 |
| Walked > 30 meters–no., % | 124 (13.1) | 12 (3) | 112 (20.3) | <0.001* | 50 (9.7) | 74 (17) | 0.001* | 31 (6.4) | 93 (19.9) | <0.001** |
| Days until first occurrence | 0 [0–5] | 3 [0–9] | 0 [0–5] | 0.03* | 0 [0–5] | 0 [0–3] | 0.19 | 3 [0–5] | 0 [0–3] | 0.04** |
| At ICU discharge | | | | | | | | | | |
| Out of bed–no., % | 526 (63.3) | 157 (40.7) | 369 (82.9) | <0.001* | 233 (52) | 293 (76.5) | <0.001* | 197 (45.1) | 329 (83.5) | <0.001* |
| Walked any distance–no., % | 335 (40.3) | 75 (19.4) | 260 (58.4) | <0.001* | 138 (30.8) | 197 (51.4) | <0.001* | 109 (24.9) | 226 (57.4) | <0.001* |
| 1–15 meters | 91 (11) | 30 (7.8) | 61 (13.7) | | 46 (10.3) | 45 (11.7) | | 48 (11) | 43 (10.9) | |
| 15–30 meters | 74 (8.9) | 24 (6.2) | 50 (11.2) | <0.001* | 35 (7.8) | 39 (10.2) | <0.001* | 28 (6.4) | 46 (11.7) | <0.001* |
| > 30 meters | 170 (20.5) | 21 (5.4) | 149 (33.5) | | 57 (12.7) | 113 (29.5) | | 33 (7.6) | 137 (34.8) | |

Data are median and interquartile range [quartile 25%—quartile 75%] or n (%). Percentages may not total 100 because of rounding.

*Definition of abbreviations*: SAPS = simplified acute physiology score; ICU = intensive care unit.

*Statistically significant difference between groups.

including premorbid conditions and prolonged periods of bed rest [38–41]. The skeletal muscle wasting in critical illness has been shown to decrease an average rate of 10–20% from ICU admission to day 7 [15, 17]. There are other important risk factors for ICU-AW, such as the severity of illness and age [38, 41]. The presence of comorbidities may predispose to a severity of muscle weakness, and ICU-AW contributes to lower physical functioning and poorer quality of life after ICU admission [42].

A study conducted in COVID-19 survivors requiring MV support showed that the majority of patients were not functionally independent at ICU discharge [43]. A need for additional use of resources to support patients during their recovery process has been observed as a

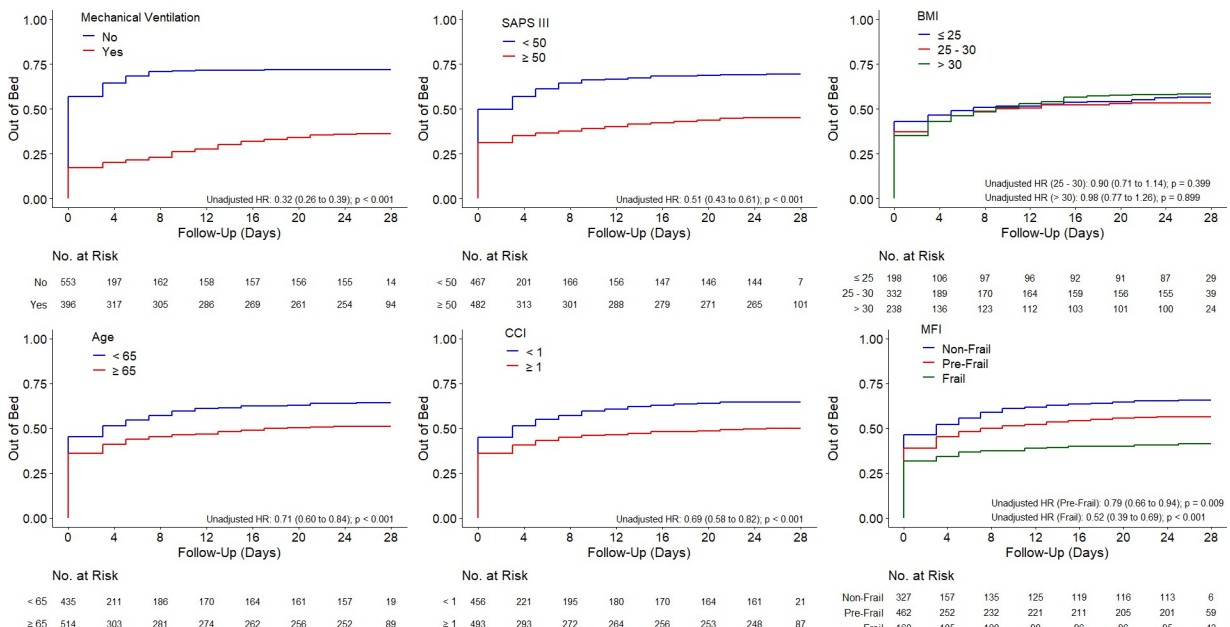

**Fig 2. Kaplan-Meier curves of time until the first day the patient got out of bed.** *Definition of abbreviations*: HR = Hazard ratio; MV = mechanical ventilation; SAPS III = simplified acute physiology score; BMI = body mass index; CCI = Charlson comorbidity index; MFI = Modified Frailty Index. Unadjusted hazard ratio calculated with a Cox proportional hazard model. To account for the competing risk of death, patients who died without achieving the event of interest were assigned the worst time possible. **a)** mechanical ventilation, with groups required (MV = yes) or not required (MV = no); **b)** simplified acute physiology score (SAPS III score); **c)** body mass index (BMI) calculated by weight in kilograms divided by the square of the height in meters (Kg/m²), categorized into groups normal or healthy weight (BMI ≤25·0), overweight (BMI = 25·0–29·9), and obese (BMI ≥30·0); **d)** age (<65 or ≥65); **e)** Charlson comorbidity index with groups <1 or ≥ 1; and **f)** Modified Frailty Index at the admission, with groups non-frail (MFI = 0), pre-frail (MFI = 1–2) and frail (MFI ≥3).

consequence of discharge home with low walking distance. It is important to clarify that patients with a low rate of independence for physical function and activities of daily living (ADLs) will demand significant follow-up medical care and rehabilitation after home discharge [43]. The COVID-19 circumstance has placed constraints on access to in-hospital rehabilitation. These findings underscore the need for prospective studies to ascertain the short-term and long-term sequelae in COVID-19 survivors [43]. In our study, one in five patients was able to walk more than 30 meters at ICU discharge. However, this rate dramatically drops to only one in twenty patients in those requiring MV support. The majority of survivors (69%) were able to ambulate > 30 meters at ICU discharge after being submitted to early activities during ICU stay as reported by Bailey and colleagues [44]. In order to reduce the consequences of bed rest, rehabilitation and early mobilization activities have been shown to be feasible and safe in ICU, preventing or treating neuromuscular complications of critical illness [44–46]. Therefore, a few years ago we implemented an early mobilization protocol aiming to reduce the bed rest consequences. However, patients' clinical condition restricted them to in-bed exercises, consequently affecting their physical function as well as their ability to walk more than 30 meters at ICU discharge. This highlights the need for continuous rehabilitation care for these patients.

Our study has limitations. First, this is a single-center retrospective observational study. However, due to the significant number of patients enrolled, our findings can contribute to identifying potential risk factors that impact on mobility status of COVID-19 patients admitted to the ICU. Second, the time frame established was limited to follow-up only during ICU stay; we did not follow the patients' mobility status until hospital discharge. Third, patients

with missing data were excluded from the analysis due to the impossibility of establishing a primary endpoint, which can be considered a potential biased sample when we analyze patients admitted to the ICU.

## Conclusion

In this single-center cohort study, the reduction of mobility level was observed in less than half of mechanically ventilated critically ill COVID-19 patients. The long-term consequences of COVID-19 on mobility level of patients requiring MV support should be investigated in future studies.

## Supporting information

**S1 Table. Rate of missing data.** Definition of abbreviations: SAPS: simplified acute physiology score; SOFA = sequential organ failure assessment; ICU = intensive care unit; COPD = chronic obstructive pulmonary disease; $PaO_2$ = partial pressure of oxygen; $FiO_2$ = fraction of inspired oxygen; $PaCO_2$ = partial pressure of carbon dioxide; ECMO = extracorporeal membrane oxygenation.
(DOCX)

**S2 Table. Organ support during ICU stay.** Data are median and interquartile range (quartile 25%—quartile 75%) or n (%). Percentages may not total 100 because of rounding. *Definition of abbreviations*: PMI = perme mobility index; ICU = intensive care unit; NMBA = neuromuscular blockade; ECMO = extracorporeal membrane oxygenation. *The Perme Mobility Index (PMI) is calculated by the difference between the total Perme Score at ICU discharge and the total Perme Score at ICU admission, divided by the ICU length of stay (ICU LOS) [PMI = ΔPerme Score (*ICU discharge–ICU admission*) / ICU LOS]. The result is a dimensionless number and it can be either positive or negative. Positive values are associated with patients that improve the mobility status during ICU stay, whereas negative values are associated with patients that decrease mobility status during ICU stay.
(DOCX)

**S3 Table. Perme description in the included patients.** Data are median and interquartile range (quartile 25%—quartile 75%) or n (%). Percentages may not total 100 because of rounding. *Definition of abbreviations*: PMI = perme mobility index; ICU = intensive care unit. *The Perme Mobility Index (PMI) is calculated by the difference between the total Perme Score at ICU discharge and the total Perme Score at ICU admission, divided by the ICU length of stay (ICU LOS) [PMI = Δ Perme Score (*ICU discharge–ICU admission*) / ICU LOS]. The result is a dimensionless number and it can be either positive or negative. Positive values are associated with patients that improve the mobility status during ICU stay, whereas negative values are associated with patients that decrease mobility status during ICU stay. †Perme ICU mobility score range from 0 to 32, with higher scores indicating better mobility level.
(DOCX)

**S4 Table. Multicollinearity and linearity assumption in the final model.** *Definition of abbreviations*: GVIF = generalized variance-inflation factor; df = degrees of freedom; SAPS = simplified acute physiology score; SOFA = Sequential Organ Failure Assessment; ECMO = extracorporeal membrane oxygenation; NMBA = neuromuscular blockade; ICU = intensive care unit.
(DOCX)

**S5 Table. Degree of mobilization according to baseline status.** Data are median and inter-quartile range (quartile 25%—quartile 75%) or n (%). Percentages may not total 100 because of rounding. *Definition of abbreviations*: ICU = intensive care unit. *Charlson comorbidity index range from 0 to 5 for each comorbidity, with score of zero indicating that no comorbidities were found. The higher the score, the more likely the predicted outcome will result in mortality or higher resource use. [†]The body-mass index (BMI) is calculated by weight in kilograms divided by the square of the height in meters ($Kg/m^2$). The categories are the same for men and women of all body types and ages, as follows: below 18.5 –underweight, 18.5–24.9 –normal or healthy weight, 25.0–29.9 –overweight, and 30.0 and above–obese. [‡]Modified Frailty Index– categorized frailty using MFI values into non-frail (MFI = 0), pre-frail (MFI = 1–2) or frail (MFI ≥ 3).
(DOCX)

**S1 Fig. Perme score in the first five days and at discharge.** Boxes represent median and inter-quartile range. Whiskers extend 1.5 times the interquartile range beyond the first and third quartiles per the conventional Tukey method. Transparent circles beyond the whiskers represent outliers. Filled circles represent mean values. *Perme ICU mobility score range from 0 to 32, with higher scores indicating better mobility level.
(DOCX)

**S2 Fig. Improvement in mobility over time.** *Definition of abbreviations*: ICU = intensive care unit; SAPS III = simplified acute physiology score. The SAPS III score ranges from 0 to 217, with higher scores indicating more severe illness and higher risk of death.
(DOCX)

**S3 Fig. Improvement in mobility over time in specific subgroups.** *Scores on SAPS III range from 0 to 217, with higher scores indicating more severe illness and higher risk of death. [†]Charlson comorbidity index range from 0 to 5 for each comorbidity, with score of zero indicating that no comorbidities were found. The higher the score, the more likely the predicted outcome will result in mortality or higher resource use. body mass index (BMI) calculated by weight in kilograms divided by the square of the height in meters ($Kg/m^2$), categorized into groups normal or healthy weight (BMI ≤ 25.0), overweight (BMI = 25.0–29.9), and obese (BMI ≥ 30.0). [§]Modified Frailty Index–categorized frailty using MFI values into non-frail (MFI = 0), pre-frail (MFI = 1–2) or frail (MFI ≥ 3).
(DOCX)

**S4 Fig. Kaplan-Meier curves of time until the first walking.** Unadjusted hazard ratio (HR) calculated with a Cox proportional hazard model. To account for the competing risk of death, patients who died without achieving the event of interest were assigned the worst time possible. **a)** mechanical ventilation (MV), with groups required (MV = yes) or not required (MV = no); **b)** simplified acute physiology score (SAPS III score); **c)** body mass index (BMI) calculated by weight in kilograms divided by the square of the height in meters ($Kg/m^2$), categorized into groups normal or healthy weight (BMI ≤ 25.0), overweight (BMI = 25.0–29.9), and obese (BMI ≥ 30.0); **d)** age (< 65 or ≥ 65); **e)** Charlson comorbidity index (CCI) with groups <1 or ≥ 1; and **f)** Modified Frailty Index (MFI) at the admission, with groups non-frail (MFI = 0), pre-frail (MFI = 1–2) and fral (MFI ≥ 3).
(DOCX)

**S5 Fig. Kaplan-Meier curves of time until the first walking of more than 30 meters.** Unadjusted hazard ratio (HR) calculated with a Cox proportional hazard model. To account for the competing risk of death, patients who died without achieving the event of interest were

assigned the worst time possible. **a)** mechanical ventilation (MV), with groups required (MV = yes) or not required (MV = no); **b)** simplified acute physiology score (SAPS III score); **c)** body mass index (BMI) calculated by weight in kilograms divided by the square of the height in meters (Kg/m$^2$), categorized into groups normal or healthy weight (BMI ≤ 25.0), overweight (BMI = 25.0–29.9), and obese (BMI ≥ 30.0); **d)** age (< 65 or ≥ 65); **e)** Charlson comorbidity index (CCI) with groups <1 or ≥ 1; and **f)** Modified Frailty Index (MFI) at the admission, with groups non-frail (MFI = 0), pre-frail (MFI = 1–2) and frail (MFI ≥ 3). (DOCX)

**S1 Dataset.**
(XLSX)

## Acknowledgments

We thank all staff members of the multidisciplinary team of Hospital Israelita Albert Einstein who managed patients during the SARS-CoV-2 outbreak. The authors thank Helena Spalic for proofreading this manuscript and Andreia Pardini for research support.

## Author Contributions

**Conceptualization:** Ricardo Kenji Nawa, Karina Tavares Timenetsky.

**Data curation:** Ary Serpa Neto, Thais Dias Midega.

**Formal analysis:** Ary Serpa Neto.

**Investigation:** Ricardo Kenji Nawa, Ary Serpa Neto, Karina Tavares Timenetsky.

**Methodology:** Ricardo Kenji Nawa, Ary Serpa Neto, Karina Tavares Timenetsky.

**Project administration:** Ricardo Kenji Nawa, Karina Tavares Timenetsky.

**Supervision:** Ricardo Kenji Nawa, Karina Tavares Timenetsky.

**Validation:** Ricardo Kenji Nawa, Ary Serpa Neto, Thiago Domingos Corrêa, Karina Tavares Timenetsky.

**Writing – original draft:** Ricardo Kenji Nawa, Ary Serpa Neto, Ana Carolina Lazarin, Ana Kelen da Silva, Camila Nascimento, Thais Dias Midega, Raquel Afonso Caserta Eid, Thiago Domingos Corrêa, Karina Tavares Timenetsky.

**Writing – review & editing:** Ricardo Kenji Nawa, Ary Serpa Neto, Thais Dias Midega, Thiago Domingos Corrêa, Karina Tavares Timenetsky.

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
