## [Decision Letter · Decision Letter 0]

5 Jul 2022

PONE-D-21-40723Degree of mobilization in critically ill patients with COVID-19: a single center, retrospective cohort study.PLOS ONE

Dear Dr. Nawa,

Thank you for submitting your manuscript to PLOS ONE. After careful consideration, we feel that it has merit but does not fully meet PLOS ONE’s publication criteria as it currently stands. Therefore, we invite you to submit a revised version of the manuscript that addresses the points raised during the review process.

We look forward to receiving your revised manuscript.

Kind regards,

Martin Kieninger

Academic Editor

PLOS ONE

Journal Requirements:

2. Please amend your current ethics statement to include the information you have given in your Methods section regarding (1) the full name of the ethics committee/institutional review board that approved your specific study, (2) the approval number given, and (3) the statement that informed consent was waived.

Reviewers' comments:

Reviewer's Responses to Questions

**Comments to the Author**

1. Is the manuscript technically sound, and do the data support the conclusions?

Reviewer #1: Yes

Reviewer #2: Yes

2. Has the statistical analysis been performed appropriately and rigorously? 

Reviewer #1: Yes

Reviewer #2: Yes

3. Have the authors made all data underlying the findings in their manuscript fully available?

Reviewer #1: Yes

Reviewer #2: Yes

4. Is the manuscript presented in an intelligible fashion and written in standard English?

Reviewer #1: Yes

Reviewer #2: Yes

5. Review Comments to the Author

Reviewer #1: Title of the manuscript doesn't explore research work. The abstract section must be re-written in order to explain and explore research idea more obviously. Patients age is very wide range the author didn't consider aging acquired weakness.

Reviewer #2: The authors explored the degree of mobilization in patients with severe COVID-19 who required mechanical ventilation (MV). The results indicated that Less than 50% of patients who underwent MV would reduce mobility level. However, there are still some issues that might need to be clarified.

MAJOR COMMENT

Introduction

The whole introduction is well-written and easy to follow. I recommend adding some studies that reported the poor outcomes of low mobility level or late mobilization in a hospital to emphasize the importance of conducting the current study.

Methods

1. Based on the title, I assume that this study only included patients with “severe” COVID-19. How did you determine the severity? Please define it in the inclusion criteria.

2. Please describe the general physiotherapy program.

3. I was wondering what the Minimal clinically important difference (MCID) of Perme Score or Perme Mobility Index (PMI) is. Was there any previous study considering “PMI>0” as improvement in mobility?

4. The sample size was quite large. Why did the authors still decide to report the median and interquartile range rather than the mean and standard deviation?

5. Did any participant receive vaccine before admission? I consider the vaccine injection as an important confounding factor.

6. What if the dependent variable is “high Perme score” and “low Perme score” in the logistic regression rather than the improvement in Perme score? Have you ever considered dividing the Perme scores at discharge into “high” and “low”? High Perme score at baseline might cause less improvement at discharge.

7. Did you ever perform a Chi-square test to analyze if the use of MV influenced improvement in PMI?

Discussion

Did any previous studies reveal a lower Perme score at admission would result in less improvement in mobility? I think this factor is worthy to be discussed.

MINOR COMMENT

Abstract

1. Please define NIV when first using it.

Results

1. What does “limitation of therapy orders” mean?

6. PLOS authors have the option to publish the peer review history of their article (what does this mean?). If published, this will include your full peer review and any attached files.

Reviewer #1: No

Reviewer #2: No

---

## [Author Response · Author response to Decision Letter 0]

11 Jul 2022

July 11, 2022

PONE-D-21-40723

Title: “Degree of mobilization in critically ill patients with COVID-19: a single center, retrospective cohort study.”

Dear Academic Editor Martin Kieninger,

Thank you for the opportunity to revise and resubmit our manuscript entitled: “Degree of mobilization in critically ill patients with COVID-19: a single center, retrospective cohort study”. We are now changing the title of the manuscript for: “Analysis of mobility level of COVID-19 patients undergoing mechanical ventilation support: a single center, retrospective cohort study”.

We are grateful for the reviewers’ time and effort. After carefully considering the suggestions and comments, we have made some changes. Our point-by-point responses to the reviewer’s comments are given below. We are very grateful for your positive and constructive feedback and we hope that our responses will answer all your queries. All changes are highlighted in the revised manuscript and described as the following in the form of point-by-point replies to the reviewer comments.

We hope that you will consider this revised version of the manuscript is favorably for publication at “PLoS ONE” as a brief report of original research.

Kind regards.

Ricardo Kenji Nawa, PT MSc PhD

Department of Critical Care Medicine

Hospital Israelita Albert Einstein

Avenida Albert Einstein, no. 627 / 701, 5th floor

ZIP: 05652- 900 – São Paulo, SP / Brazil.

E-mail: ricardo.nawa@einstein.br

Comments to the Author

Please use the space provided to explain your answers to the questions above. You may also include additional comments for the author, including concerns about dual publication, research ethics, or publication ethics. (Please upload your review as an attachment if it exceeds 20,000 characters).

Reviewer #1

Title of the manuscript doesn't explore research work. The abstract section must be re-written in order to explain and explore research idea more obviously. Patients age is very wide range the author didn't consider aging acquired weakness.

Thank you very much for all your suggestions and comments. We really appreciate the time spent revising our manuscript. The authors decided to change the original title of the manuscript for: “Analysis of mobility level of COVID-19 patients undergoing to mechanical ventilation support: a single center, retrospective cohort study”.

We also revised the entire abstract section, rewriting it, in order to better explore the research idea and findings, as suggested. We hope to have made proper adjustments needed of the content to readers better understand.

Reviewer #2

The authors explored the degree of mobilization in patients with severe COVID-19 who required mechanical ventilation (MV). The results indicated that Less than 50% of patients who underwent MV would reduce mobility level. However, there are still some issues that might need to be clarified.

The authors appreciate the suggestions and comments provided along the manuscript. After a careful revision of the entire manuscript we hope all “issues” have been adequately reported as suggested.

MAJOR COMMENT

Introduction

The whole introduction is well-written and easy to follow. I recommend adding some studies that reported the poor outcomes of low mobility level or late mobilization in a hospital to emphasize the importance of conducting the current study.

The authors agree with the suggestion to add some studies in order to report mobility level, physical and clinical outcomes after episode of critical illness. We included three studies into “introduction” section, as follows: “The consequences are frequently observed among ICU survivors, with impairment physical and psychological recovery [Boelens et al. (2022)]. Early mobilization and rehabilitation have been shown to be effective in enhancing the recovery of critically ill patients, but more large-scale, multicenter randomized controlled trials are required to further confirm these findings [Monsees et al. (2022)]. Although the present available evidence of early rehabilitation still remains inconsistent; on the overall evidence rehabilitation interventions should not be delayed [Wang et al. (2020)].”

1. Boelens YFN, Melchers M, van Zanten ARH. Poor physical recovery after critical illness: incidence, features, risk factors, pathophysiology, and evidence-based therapies. Curr Opin Crit Care. 2022 Jul 6. doi: 10.1097/MCC.0000000000000955. Epub ahead of print. PMID: 35796071.

2. Monsees J, Moore Z, Patton D, Watson C, Nugent L, Avsar P, O'Connor T. A systematic review of the effect of early mobilization on length of stay for adults in the intensive care unit. Nurs Crit Care. 2022 Jun 1. doi: 10.1111/nicc.12785. Epub ahead of print. PMID: 35649531.

3. Wang J, Ren D, Liu Y, Wang Y, Zhang B, Xiao Q. Effects of early mobilization on the prognosis of critically ill patients: A systematic review and meta-analysis. Int J Nurs Stud. 2020 Oct;110:103708. doi: 10.1016/j.ijnurstu.2020.103708. Epub 2020 Jul 11. PMID: 32736250.

Methods

1. Based on the title, I assume that this study only included patients with “severe” COVID-19. How did you determine the severity? Please define it in the inclusion criteria.

After read your comment, the authors considered the topic of “inclusion criteria” inconsistent. We would like to clarify that not only “severe” cases of COVID-19 were included in the present study. We included all patients with COVID-19 diagnosis (e.g., mild, moderate, and severe) cases that required ICU admission.

We decided to provide additional information about the inclusion criteria adopted, as follows: “We considered all ICU admissions of COVID-19 diagnosis (e.g., mild, moderate, and severe) during the study period eligible for inclusion. The following inclusion criteria were used: 1) age ≥ 18 years; and 2) confirmed diagnosis of COVID-19 by reverse transcription–polymerase chain reaction (RT-PCR) for SARS-CoV-2 [20]. We excluded patients with missing core data, defined as the use of MV during ICU stay and report of mobility status at ICU admission and/or discharge”. We hope this modification meets your suggestion for further clarification regarding the inclusion criteria adopted, improving readers understand.

2. Please describe the general physiotherapy program.

The general physiotherapy program at Hospital Israelita Albert Einstein, consists in five levels of activity therapy. The advance through levels 1 through 5, is based on patients’ ability to execute tasks gradually complex, followed by increase of muscle strength and less dependence to execute the proposed activities and exercises. There is a gradual progression on patients’ physical function status, as briefly presented below:

• Level “1”, is considered when patients were unconscious and restrict to bed. Only passive range of motion therapy, stretching, and positioning is administered to all upper and lower extremity joints. The use of equipment such as neuromuscular electrical stimulation (NMES), and passive mode of cycle ergometer can be used as part of physiotherapy interventions.

• Level “2”, is considered when patients are able to sit at side of bed. They progress to active-assistive and active range of motion exercise as they are alert and able to advance their participation during physiotherapy interventions.

• Level “3”, patients are able to stand and, in some cases, transfer to edge of bed. As patients progressed, the activities will be more complex, increasingly focused on functional activities, requiring more attention, coordination, and performance.

• Level “4”, will be considered for those patients who are able to execute pre-gait activities (e.g., forward and lateral weight shifting, marching in place) and ambulation initiate for short distances. They gradually require less assistance to ambulate for greater distances.

• Level “5”, patients are ambulating without any assistance or use of gait devices. The focus at this stage is prepare the patient to be discharged home, educating them how to maintain the functional independence, and activities of daily living.

3. I was wondering what the Minimal clinically important difference (MCID) of Perme Score or Perme Mobility Index (PMI) is. Was there any previous study considering “PMI>0” as improvement in mobility?

Thanks for your comment. The authors would like to clarify that the minimal clinically important difference (MCID) of Perme Score was first established by Wilches Luna et at. – “Wilches Luna EC, de Oliveira AS, Perme C, Gastaldi AC. Spanish version of the Perme Intensive Care Unit Mobility Score: Minimal detectable change and responsiveness. Physiother Res Int. 2021 Jan;26(1):e1875. doi: 10.1002/pri.1875. Epub 2020 Sep 14. PMID: 32926503.”

This single-center study conducted with ICU patients, determined the value of 1.36 points of the Perme Score, to be able to detect changes on patients’ mobility status. This value can be considered questionable, due to the narrow value presented, once the Perme Score total points range from "0" up to "32" points. However, to our knowledge, this is the only available publication establishing the MCID for the Perme Score.

Additionally, our research group recently published a study entitled: “The Perme Mobility Index: A new concept to assess mobility level in patients with coronavirus (COVID-19) infection” – “Timenetsky KT, Serpa Neto A, Lazarin AC, Pardini A, Moreira CRS, Corrêa TD, Caserta Eid RA, Nawa RK. The Perme Mobility Index: A new concept to assess mobility level in patients with coronavirus (COVID-19) infection. PLoS One. 2021 Apr 21;16(4):e0250180. doi: 10.1371/journal.pone.0250180. PMID: 33882081; PMCID: PMC8059854.”

This new concept of the “Perme Mobility Index (PMI)” established values of “PMI>0” as improvement in mobility status, once it is calculated, as presented below: “[PMI = ΔPerme Score (ICU discharge – ICU admission)/ICU length of stay]”. Based on the PMI, patients can be divided into two groups: “Improved mobility status” (PMI > 0) and “Not improved mobility status” (PMI ≤ 0). Thus, the “positive” values (e.g., PMI > 0) must be interpreted with Perme Score at ICU discharge greater than the Perme Score at ICU admission. The “negative” values (e.g., PMI ≤ 0) must be interpreted with Perme Score at ICU discharge lower than the Perme Score at ICU admission.

4. The sample size was quite large. Why did the authors still decide to report the median and interquartile range rather than the mean and standard deviation?

We appreciate the reviewer’s comment. Indeed, the sample size of the present study can be considered “quite large” (n = 949). The results are reported as median and interquartile range, because according to our statistical analysis, the normality was assessed by the Kolmogorov-Smirnov test. We included this information for readers better understanding, as follows: “Continuous variables are presented as median and interquartile range (IQR), and categorical variables as absolute and relative frequencies. Normality was assessed by the Kolmogorov-Smirnov test.”

5. Did any participant receive vaccine before admission? I consider the vaccine injection as an important confounding factor.

Thanks for your comment. This study included patients between “February 2020 to February 2021”, as reported at the first paragraph of the “results” section, as follows: “From February 2020 to February 2021, 1,297 patients with confirmed COVID-19 were admitted to the ICU, of which a total of 949 (73.1%) met the inclusion criteria and were included in subsequent analysis”.

According to the Ministry of Health of Brazil, the Brazilian National Immunization Program, started the COVID-19 vaccination campaign in Brazil on January 17, 2021 (https://www.gov.br/anvisa/pt-br/english), which means that possibly none or just a few number of included patients may have been vaccinated by the time of hospital admission.

We did not collected data about patients’ vaccination rate of included sample of this study. As mentioned before, perhaps a very small number of patients may have received the first dose of COVID-19 vaccine. However, once we do not have this data, we cannot guarantee if some of them received or not the vaccine prior to hospital admission.

6. What if the dependent variable is “high Perme score” and “low Perme score” in the logistic regression rather than the improvement in Perme score? Have you ever considered dividing the Perme scores at discharge into “high” and “low”? High Perme score at baseline might cause less improvement at discharge.

Thanks for your comment. The authors considered this suggestion a good viewpoint. In fact, the idea to consider the dependent variable as “high and low Perme score” in the logistic regression analysis rather than the improvement or not of the Perme Mobility Index (PMI) is supported by logical thinking. However, it is important to cite emphasize that we have never thought divide the Perme score into two “high” and “low” scores, once we do not know how high the total Perme Score needs to be in order to be considered a “high” score. The same occur for “low” scores, we do not know, and it is still not stablished in the literature, how low the total Perme Score need to be in order to be considered a "low" score.

Maybe a cutoff point on the total Perme Score, should be establish and explored in future studies, in order to help the interpretation of clinical use of the Perme Score, as well as to provide the possibility to classify patients’ mobility status as “high” and “low” scores, once the total score ranges from “0” to “32” points.

The authors agree that “high” Perme score at baseline might cause less improvement at discharge moment. However, this hypothesis needs to be confirmed in future studies, otherwise we will be just inferring a possible conclusion that might be right or not.

7. Did you ever perform a Chi-square test to analyze if the use of MV influenced improvement in PMI?

Thank you for your comment. The authors are aware that “Chi-square” test could be used to compare observed results with expected results, determining if a difference between observed and expected data is due to chance, or if it is due to a relationship between the variables we are studying. But we have never performed this analysis. In this case, we could have checked if the use of mechanical ventilation may have influenced the improvement of the PMI, as suggested. However, once this analysis was not part of the original statistical analysis plan, we did not perform this analysis in this study.

Discussion

Did any previous studies reveal a lower Perme score at admission would result in less improvement in mobility? I think this factor is worthy to be discussed.

Thank you for your comment. We did an extensively search of the literature to make sure that any new study could have been published revealing this association of a lower Perme Score at admission would result in less improvement in patients’ mobility status. However, we did not find any new publication since our study entitled: “The Perme Mobility Index: A new concept to assess mobility level in patients with coronavirus (COVID-19) infection” – “Timenetsky KT, Serpa Neto A, Lazarin AC, Pardini A, Moreira CRS, Corrêa TD, Caserta Eid RA, Nawa RK. The Perme Mobility Index: A new concept to assess mobility level in patients with coronavirus (COVID-19) infection. PLoS One. 2021 Apr 21;16(4):e0250180. doi: 10.1371/journal.pone.0250180. PMID: 33882081; PMCID: PMC8059854.”

In this retrospective single-center cohort study, the mobility level, measured by the Perme Score, in critically ill COVID-19 patients was low at ICU admission; however, most patients improved their mobility level during ICU stay. We also reported the risk factors associated with mobility level were: 1) age; 2) comorbidities; and 3) use of renal replacement therapy.

We decided to add a sentence in the discussion section in order to improve readers understanding about the mobility level measured at ICU admission, as follows: “A retrospective single-center cohort study, the mobility level of critically ill COVID-19 patients measured by the Perme Score, was low at ICU admission; however, most patients improved their mobility level during ICU stay [18]. This study also reported the risk factors associated with mobility level were age, comorbidities, and use of renal replacement therapy [18].”

MINOR COMMENT

Abstract

1. Please define NIV when first using it.

Thanks for your suggestion. We added the definition for “NIV” and we decided to remove the use of the abbreviation, as follows: “…while non-invasive ventilation and vasopressor…”

Results

1. What does “limitation of therapy orders” mean?

Thanks for your comment. We used an inappropriate term and we have replaced it for “palliative care status”, as follows: “…presence of frailty, palliative care status, use of MV after the first hour of ICU admission…”

---

## [Decision Letter · Decision Letter 1]

19 Jul 2022

“Analysis of mobility level of COVID-19 patients undergoing mechanical ventilation support: a single center, retrospective cohort study”.

PONE-D-21-40723R1

Dear Dr. Nawa,

We’re pleased to inform you that your manuscript has been judged scientifically suitable for publication and will be formally accepted for publication once it meets all outstanding technical requirements.

Kind regards,

Martin Kieninger

Academic Editor

PLOS ONE

Additional Editor Comments (optional):

Reviewers' comments:

Reviewer's Responses to Questions

**Comments to the Author**

1. If the authors have adequately addressed your comments raised in a previous round of review and you feel that this manuscript is now acceptable for publication, you may indicate that here to bypass the “Comments to the Author” section, enter your conflict of interest statement in the “Confidential to Editor” section, and submit your "Accept" recommendation.

Reviewer #1: All comments have been addressed

Reviewer #2: (No Response)

2. Is the manuscript technically sound, and do the data support the conclusions?

Reviewer #1: Yes

Reviewer #2: Yes

3. Has the statistical analysis been performed appropriately and rigorously? 

Reviewer #1: Yes

Reviewer #2: Yes

4. Have the authors made all data underlying the findings in their manuscript fully available?

Reviewer #1: Yes

Reviewer #2: Yes

5. Is the manuscript presented in an intelligible fashion and written in standard English?

Reviewer #1: Yes

Reviewer #2: Yes

6. Review Comments to the Author

Reviewer #1: Thanks for your efforts in correcting the previously mentioned notes. The manuscript seeks a serious problem interfacing health care practitioners

Reviewer #2: (No Response)

7. PLOS authors have the option to publish the peer review history of their article (what does this mean?). If published, this will include your full peer review and any attached files.

Reviewer #1: **Yes: **Tamer I. Abo Elyazed, Ass.prof of Physical Therapy for internal medicine,Beni-Suef University

Reviewer #2: **Yes: **FU-LIEN WU

---

## [Editor Report · Acceptance letter]

22 Jul 2022

PONE-D-21-40723R1 

Analysis of mobility level of COVID-19 patients undergoing mechanical ventilation support: a single center, retrospective cohort study. 

Dear Dr. Nawa:

I'm pleased to inform you that your manuscript has been deemed suitable for publication in PLOS ONE. Congratulations! Your manuscript is now with our production department. 

Kind regards, 

on behalf of

Dr. Martin Kieninger 

Academic Editor

PLOS ONE